# Are Gluten-Free Diets More Nutritious? An Evaluation of Self-Selected and Recommended Gluten-Free and Gluten-Containing Dietary Patterns

**DOI:** 10.3390/nu10121881

**Published:** 2018-12-03

**Authors:** Amy Taetzsch, Sai Krupa Das, Carrie Brown, Amy Krauss, Rachel E. Silver, Susan B. Roberts

**Affiliations:** Jean Mayer USDA Human Nutrition Research Center on Aging, Tufts University, 711 Washington Street, Boston, MA 02111, USA; amy.taetzsch@tufts.edu (A.T.); Sai.Das@tufts.edu (S.K.D.); cabrown23@gmail.com (C.B.); akrauss0@gmail.com (A.K.); Rachel.Silver@Tufts.edu (R.E.S.)

**Keywords:** gluten, nutrient composition, energy intake, dietary fiber, folate, MyPlate

## Abstract

Gluten-free (GF) eating patterns are frequently perceived to be healthier than gluten-containing (GC) ones, but there has been very little research to evaluate this viewpoint. The effect of GF eating patterns on dietary composition was assessed using two independent approaches. One approach compared macronutrients and typical shortfall nutrients between MyPlate example menus developed with either GC or equivalent GF foods. In this analysis, the GF menus were significantly lower in protein, magnesium, potassium, vitamin E, folate, and sodium (*p* = 0.002–0.03), with suggestive trends towards lower calcium and higher fat (*p* = 0.06–0.08). The second approach was a meta-analysis of seven studies comparing information on the nutrient intakes of adults with celiac disease following a GF diet with control subjects eating a GC diet, and differences were evaluated using paired *t*-tests or Wilcoxon Signed rank tests. In this analysis, consuming a GF diet was associated with higher energy and fat intakes, and lower fiber and folate intakes compared to controls (*p* < 0.001 to *p* = 0.03). After adjusting for heterogeneity and accounting for the large mean effect size (−0.88 ± 0.09), the lower fiber remained significant (*p* < 0.001). These combined analyses indicate that GF diets are not nutritionally superior except for sodium, and in several respects are actually worse.

## 1. Introduction

Gluten-free (GF) eating patterns have become a mainstream phenomenon during recent years, and nearly one-third of Americans report having attempted to eliminate or reduce the amount of dietary gluten they consume [1]. ‘Good health’ is the primary reason given for consuming a GF diet [1], despite the fact that only 6% of the U.S. population has non-celiac gluten sensitivity and less than 1% has celiac disease [2,3]. Celiac disease is an autoimmune disorder against the protein, gluten; the only therapy is life-long consumption of a GF diet [3]. However, there is very little information available on the nutritional quality of GF diets, and therefore it is not known whether GF diets are actually healthier for individuals without a medical need to restrict gluten [4,5,6,7]. 

A small number of reports on the nutritional composition of foods based on package labels have indicated the composition of GF products are similar in energy, macronutrients, and/or micronutrients to gluten-containing (GC) products, with the exception of slightly lower protein content [8,9]. However, these analyses have limited translation to the nutrient composition of a GF diet, as they do not account for the nutrient consumption of people following a GF diet or what the target composition of therapeutically recommended dietary patterns would be if GF foods were substituted for GC foods in an otherwise healthy diet.

The objective of this study was to provide additional information on the healthfulness of GF eating patterns and to test the hypothesis that both self-selected and recommended eating patterns are less nutritious when GF versus GC dietary patterns are followed. 

## 2. Materials and Methods 

Two studies were conducted to examine differences in the nutrient quality of GF and GC diets. Study 1 examined the nutrient profiles of healthy menus based on the principles of the Dietary Guidelines [10] implemented as GF versus GC versions of the published example menus provided by the United States Department of Agriculture (USDA) for MyPlate menus [11]. Study 2 was a literature review of the nutritional composition of self-selected GF diets compared with GC diets of matched control subjects, and used meta-analysis techniques to summarize the findings. Available data were analyzed for energy, macronutrients, and limiting micronutrients (vitamin A, vitamin D, vitamin E, vitamin C, folate, calcium, magnesium, fiber, and potassium). Iron is also a limiting nutrient for premenopausal women [10] but was not included in the analyses because the available data were not sufficient to analyze by subgroup. This study was deemed exempt from institutional review board approval under federal regulation 45 46.101 (b) Code of Federal Regulations.

### 2.1. Study 1: Analysis of Healthy Dietary Recommendations Implemented as GF and GC Menus

A theoretical analysis was conducted to examine the effect of a healthy dietary pattern with either GF or GC foods. The USDA MyPlate nutrition guide [11] was used as the healthy diet standard because there are seven reference menus for a 2000 calorie daily food pattern that could serve as a template. The nutrient compositions of these reference menus were calculated using the USDA National Nutrient Database for Standard Reference Release 27 and nutrition facts labels and served as GC control menus. To create GF equivalent menus, the GC food items in the reference menus were replaced with the most equivalent GF product identified by a registered dietitian. Oats were considered GF and included in the GF equivalent menus. Generic USDA database codes were used when available, and otherwise GF branded products were matched to the nearest weight. For example, a whole-wheat English muffin, included as a breakfast item on the GC MyPlate menu, was substituted with a brown-rice English muffin on the GF menu (see example menu, Table 1, and all menus in Appendix A).

### 2.2. Study 2: Literature Review and Meta-Analysis of Self-Reported Dietary Intakes among Individuals Consuming a GF Diet Versus a GC Diet

A literature review was conducted to identify peer-reviewed publications evaluating the dietary intake of subjects consuming a GF diet with control subjects consuming a GC diet. The purpose of the review was to compare data for energy, macronutrient, and micronutrient compositions of self-selected GF versus GC dietary regimens and conduct a meta-analysis. The following inclusion criteria were used: (1) dietary intake was obtained from food records or 24-h dietary recalls for men and women with celiac disease on a GF diet and were compared to the dietary intake of healthy controls from the same population group; (2) reported energy and at least one other nutrient of interest with mean and standard deviation or error presented for data in both groups; and (3) the study was published in English. A literature search was conducted using PubMed and Web of Science databases, and studies published between May 1994 and May 2018 were included. Relevant articles were determined by reviewing the titles and/or abstracts identified by the search. Full-text articles suspected to meet inclusion criteria were reviewed and nutrient information was extracted from eligible studies. In studies that reported results stratified by gender, data from men and women were combined and a weighted average was calculated. 

### 2.3. Statistical Analysis 

For Study 1, the seven reference menus were used to determine the average daily intake of each nutrient. The distribution of each nutrient was assessed using histograms and the Shapiro-Wilk test for normality. A *p* value < 0.05 indicated that the distribution for a specific nutrient was not normal. Based on the Shapiro-Wilk test, vitamin A, vitamin E, folate, and sodium were not normally distributed. The non-parametric Wilcox Signed-rank test was therefore used to evaluate the differences in these nutrients between the GF and GC diets. Energy, total carbohydrates, total protein, total fat, saturated fat, fiber, vitamin C, calcium, magnesium, and potassium were normally distributed. A paired *t*-test was used to evaluate the differences in these nutrients between the GF and GC diets. 

For Study 2, a meta-analysis was used to compare differences in nutrients between GF diets and GC diets across seven studies meeting our inclusion criteria. For each nutrient, standardized mean differences (effect sizes) and standard errors between the GF and GC groups from each study were calculated. For all nutrients, a fixed-effects model was initially used to derive the pooled mean difference and standard error across studies. Heterogeneity across studies was determined based on a Q-statistic < 0.05. For nutrients with significant heterogeneity, the pooled mean difference and standard error were derived using a linear mixed model. 

All analyses were conducted using Excel and SAS version 9.3 (SAS Institute, Inc., Cary, NC, USA). All statistical tests were two-sided, and a *p* value less than 0.05 was considered statistically significant.

## 3. Results

### 3.1. Study 1: Analysis of Healthy Dietary Recommendations Implemented as GF and GC Menus

Table 2 shows the mean nutrient intakes calculated for the seven-day reference MyPlate menus and GF equivalent menus. The GF version of MyPlate was significantly lower in total protein, vitamin E, folate, magnesium, potassium, and sodium compared to GC comparable menus. In addition, suggestive trends were observed for lower calcium (*p* = 0.08) and higher total fat (*p* = 0.06) in the GF menu. No other comparisons between GF and GC menus were statistically significant.

### 3.2. Study 2: Literature Review and Meta-Analysis of Self-Reported Dietary Intake among Individuals Consuming a Gluten-Free Diet Versus a Gluten-Containing Diet

The literature review identified seven articles that met the predetermined inclusion criteria for GF and equivalent GC diet data. All studies included in this analysis were observational, conducted in Europe, implemented three- to seven-day food logs to capture dietary information, had a sample size of 40 or more subjects per group, and all subjects in the GF diet group had celiac disease [12,13,14,15,16,17,18]. Energy, carbohydrates, protein, fat, fiber, vitamin D, folate, and calcium data were available in at least four of the seven included studies and were analyzed. 

The nutrient composition of medically prescribed GF diets and GC diets were examined using a fixed-effects model. Figure 1 displays the available data for macronutrient and micronutrient intake for each diet and Table 3 reports mean effect sizes. The mean total daily energy was significantly higher in GF participants compared to the GC controls (mean effect size ± standard error: 0.14 ± 0.06; *p* = 0.03). Similar results were observed for total fat intake (0.14 ± 0.06; *p* = 0.03). In addition, both fiber and folate intake were significantly lower in GF individuals compared to controls (−0.88 ± 0.09; *p* < 0.001 and −0.17 ± 0.07; *p* = 0.02, respectively). Statistical heterogeneity was a concern for all nutrients except protein and vitamin D. When heterogeneity was addressed through a random-effects model, the lower daily fiber intake within the GF diet remained statistically significant (Q = 22.28; *p* < 0.001). 

## 4. Discussion

Two different approaches were used to examine the effect of a GF diet on nutrient intakes. A comparison of calculations for MyPlate menus that included GC products or substituted GF products demonstrated that the GF menus had significantly lower amounts of several micronutrients that are defined as limiting in the U.S. diet, including magnesium, potassium, vitamin E, and folate [10]. In addition, a meta-analysis of dietary data from studies comparing the nutrient intake of individuals consuming a medically prescribed GF diet with controls consuming self-selected GC foods indicated that individuals adhering to a GF regimen consumed more calories and fat and substantially less dietary fiber and folate. The combined findings of these two different analytical approaches indicate that the nutritional quality of GF eating patterns is not healthier than GC eating patterns, and in some important respects is nutritionally inferior, even when individuals attempt to consume a healthy diet. 

The considerably lower dietary fiber intake consumed by individuals medically-selecting a GF diet is likely due to the lower consumption of grain products [19], particularly as our separate analysis of MyPlate dietary patterns indicated similar fiber intake when GF dietary grains are substituted for equivalent GC grains including oats, which were historically excluded from gluten free-diets. These findings are consistent with studies in celiac patients that report low fiber intake [16,18,20]. Dietary fiber is among the nutrients that are most lacking in the U.S. diet [10], despite strong evidence for a central role for dietary fiber in metabolic health, including glycemic control, lowering of cholesterol, satiety, and colon health [21]. The much lower dietary fiber in GF dietary patterns that include lower grain consumption is therefore likely to be detrimental to the prevention of non-communicable diseases including obesity, type 2 diabetes, and cardiovascular disease. A limitation of these findings is that none of the studies specified whether oats were included in a GF diet, however, the addition of oats would bolster the fiber content of medically selected GF diets, which was significantly lower than GC diets. We additionally observed that GF dietary patterns based on MyPlate are lower in several micronutrients (magnesium, potassium, vitamin E, and folate) identified as limiting in the U.S. diet [10]. This observation is consistent with previous reports of lower serum nutrients, such as folate, in individuals with celiac disease [17]. Although low serum folate in individuals with celiac disease can be due to malabsorption [22], our analysis indicates that low dietary folate may be a contributing factor, and, furthermore, that other limiting micronutrients may also be consumed in inadequate amounts. The one nutrient that showed beneficial changes in the GF diet was sodium, which was 330 mg lower in the GF menus compared to the GC menus. This positive change could potentially be replicated in GC foods, but at the present time is a benefit of GF foods, especially for individuals diagnosed with hypertension [23].

Although this analysis contributes to the limited body of literature related to the healthfulness of GF eating patterns, additional research is still needed because there is a lack of information on this topic generally, especially considering the widespread use of GF foods and the common public perception of them representing healthier food options. Our analysis of self-reported intake relied on data from individuals with celiac disease and, to our knowledge, there have been no studies on the nutrient composition of GF diets in individuals without a diagnosis of either celiac disease or gluten intolerance, which may influence the relative intake of different food groups [19]. In addition, analyses of GF menus are primarily conducted through assessment of nutrient databases and are limited by the lack of nutritional information for the most commonly consumed GF products.

## 5. Conclusions

Two separate analyses using conceptually different approaches indicate that GF eating patterns do not have healthier macronutrient or micronutrient profiles, with the exception of lower sodium. On the contrary, based on the results of this study GF patterns are less optimal for dietary fiber, folate, total protein, vitamin E, magnesium, and potassium. As life-long adherence to a GF diet is the only treatment for celiac disease, it is important that clinicians and registered dietitians use these emerging findings to promote healthy food choices in these patients, recognizing that less healthy diets are easier to select. In the absence of a medical indication to restrict gluten consumption [20,24,25,26], the potential for lower fiber and several limiting micronutrients highlights the potential for negative effects of GF dietary patterns on long-term health. 

## Figures and Tables

**Figure 1 nutrients-10-01881-f001:**
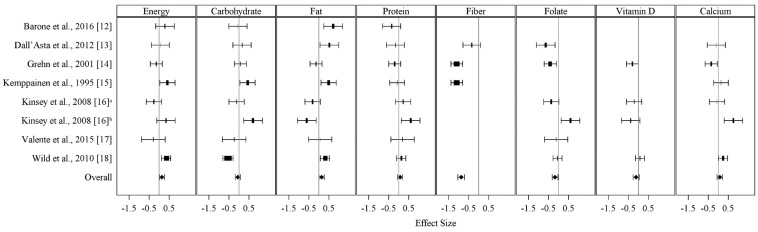
Forest plot for the difference in energy, macronutrient, and micronutrient intake by study displayed as effect size and 95% confidence limits. ^a^ age range: 19–64 years; ^b^ age range: >65 years.

**Table 1 nutrients-10-01881-t001:** Example healthy day menu.

Meal	Gluten Containing	Gluten Free
Breakfast	1 whole wheat English muffin	*1 GF brown rice English muffin*
	1 tbsp all-fruit preserves	1 tbsp all-fruit preserves
	1 hard-cooked egg	1 hard-cooked egg
	Beverage: 1 cup water, coffee, or tea	Beverage: 1 cup water, coffee, or tea
Lunch	White bean-vegetable soup:	White bean-vegetable soup:
	1 ¼ cup chunky vegetable soup with pasta	*1 ¼ cup GF vegetable noodle soup*
	½ cup white beans	½ cup white beans
	6 saltine crackers	*6 GF table crackers*
	½ cup celery sticks	½ cup celery sticks
	Beverage: 1 cup fat-free milk	Beverage: 1 cup fat-free milk
Dinner	Rigatoni with meat sauce:	Rigatoni with meat sauce:
	1 cup rigatoni pasta (2 oz dry)	*1 cup GF fusilli (2 oz dry)*
	2 ounces cooked ground beef	2 ounces cooked ground beef
	(95% lean)	(95% lean)
	2 tsp corn/canola oil (to cook beef)	2 tsp corn/canola oil (to cook beef)
	½ cup tomato sauce	½ cup tomato sauce
	3 tbsp grated parmesan cheese Spinach salad:	3 tbsp grated parmesan cheese Spinach salad:
	1 cup raw spinach leaves	1 cup raw spinach leaves
	½ cup tangerine sections	½ cup tangerine sections
	½ ounce chopped walnuts	½ ounce chopped walnuts
	4 tsp oil and vinegar dressing	4 tsp oil and vinegar dressing
	Beverage: 1 cup water, coffee, or tea	Beverage: 1 cup water, coffee, or tea
Snacks	1 cup nonfat fruit yogurt	1 cup nonfat fruit yogurt

Italicized indicates gluten-free substitute. GF: gluten-free.

**Table 2 nutrients-10-01881-t002:** Daily average nutrient composition of a healthy diet using seven-day MyPlate meal plans.

Nutrient	Gluten Free (Mean ± SD ^a^)	Gluten Containing (Mean ± SD ^a^)	*p* Value ^b^
Energy (calories)	1979 ± 211	1991 ± 209	0.31
Total Carbohydrates (g)	275 ± 53	265 ± 66	0.23
Total Protein (g)	88 ± 14	96 ± 11	0.002
Total Fat (g)	64 ± 15	62 ± 16	0.06
Saturated Fat (g)	23 ± 10	22 ± 10	0.18
Fiber (g)	30 ± 6	31 ± 7	0.84
Vitamin A (IU)	14117 ± 12285	14085 ± 12292	0.88
Vitamin C (mg)	148 ± 80	147 ± 79	0.33
Vitamin E (IU)	6 ± 2	7 ± 3	0.02
Folate (μg)	259 ± 81	340 ± 114	0.02
Calcium (mg)	1578 ± 275	1639 ± 265	0.08
Magnesium (mg)	350 ± 69	409 ± 68	0.004
Potassium (mg)	4009 ± 700	4228 ± 785	0.01
Sodium (mg)	2124 ± 597	2494 ± 765	0.03

^a^ SD: standard deviation; ^b^
*p* values calculated by a paired *t*-test (energy, total carbohydrates, total protein, total fat, saturated fat, fiber, vitamin C, calcium, magnesium, potassium) or by the Wilcoxon Sign rank test (vitamin A, vitamin E, folate, sodium).

**Table 3 nutrients-10-01881-t003:** Mean effect size for differences in energy, macronutrient, and micronutrient intake between gluten-free (GF) diets and gluten-containing (GC) diets, based on meta-analysis of seven published studies.

Nutrient	Number of Studies	Mean Effect Size (95% CI) ^a^	*p* Value ^b^	*p* Value ^c^
Energy (calories)	7	0.14 (0.02, 0.27)	0.03	0.24
Total Carbohydrates	7	−0.06 (−0.19, 0.06)	0.33	0.79
Total Protein	7	0.07 (−0.06, 0.19)	0.29	-
Total Fat	7	0.14 (0.01, 0.26)	0.03	0.44
Fiber	3	−0.88 (−1.04, −0.71)	<0.001	<0.001
Folate	5	−0.17 (−0.31, −0.03)	0.02	0.06
Calcium	5	0.08 (−0.06, 0.21)	0.26	0.60
Vitamin D	3	0.12 (−0.27, 0.03)	0.13	-

^a^ Effect Size: Difference in intake for GF diet vs. GC diet; CI: confidence interval; ^b^
*p* value for fixed-effects models; ^c^
*p* value for mixed-effects models. Mixed-effects models were conducted for nutrients with significant heterogeneity (Q-statistic < 0.05).

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
