# Peer review of "Are Gluten-Free Diets More Nutritious? An Evaluation of Self-Selected and Recommended Gluten-Free and Gluten-Containing Dietary Patterns"

_nutrients, 2018, doi:10.3390/nu10121881_

Round 1
Reviewer 1 Report
The relationship between gluten free eating patterns and "good health" is a topic of growing interest and a revision of current literature about it would be intriguing. The limit of this paper is limitated current scientific literature. In fact, there have been no studies on the nutrient provision of gluten free diet in subjects without celiac disease or with non-celiac gluten sensitivity. The paper is clearly understandable and the same is true for final message. However, it requires minor changes:
# Introduction section should be improved. Authors should include a short description of celiac disease following the data of prevalence of disease in US popolation. Moreover, thay should report that the only therapy for celiac disease patients is a life-long gluten-free diet.
The sentence from line 38 to line 40 is not clear, it should be improved.
#Material and Methods section: after " ..fiber, and potassium" add parentheses ( line 52)
# Results section: figure 2 could be improved adding a column for p values
# Conclusion section should be improved, taking into account that gluten-free diet is the only long-life diet for celiac patients. Emerged findings could be used by clinicians and especially dieticians to formulategluten-free eating patterns enriched with fiber, folate, total protein, etc...
# References section: at the end of reference 1 add parentheses.
Author Response
Point 1: # Introduction section should be improved. Authors should include a short description of celiac disease following the data of prevalence of disease in US popolation. Moreover, thay should report that the only therapy for celiac disease patients is a life-long gluten-free diet.
Response 1: Great suggestion! We added a sentence on celiac disease and the necessity of a GF diet for people with celiac disease.
Point 2: The sentence from line 38 to line 40 is not clear, it should be improved.
Response 2: Edits have been made to improve clarity of the sentence.
Point 3: #Material and Methods section: after " ..fiber, and potassium" add parentheses ( line 52)
Response 3: A parenthesis has been added, thank you.
Point 4: # Results section: figure 2 could be improved adding a column for p values
Response 4: Thank you for your suggestion. We agree, more information would be useful to present. We decided to add an additional table to show the p values and mixed effects size for both the fixed effects and mixed effects models.
Point 5: # Conclusion section should be improved, taking into account that gluten-free diet is the only long-life diet for celiac patients. Emerged findings could be used by clinicians and especially dieticians to formulategluten-free eating patterns enriched with fiber, folate, total protein, etc...
Response 5: We included the importance of clinicians using these findings to promote healthy eating in celiac patients in the conclusion.
Point 6: # References section: at the end of reference 1 add parentheses.
Response 6: This has been updated.
Thank you for your review!

Reviewer 2 Report
I have enjoyed reading of the manuscript. Well designed and documented studies, well written paper, topic highly relevant. However, I feel that one important issue needs to be solved: does the GFD in Study 1 and Study 2 include gluten-free oats or not? Gluten-free oats may increase nutritional value of a GFD via improved amino acid, fiber, mineral, and vitamin balance. In Study 2, there is necessary to emphasize inclusion/exclucion of gluten-free oats. In Study 1, it would be great to add a sub-study that calculates intake of fiber and other mentioned nutrients, minerals and vitamins.
Author Response
Point 1: I have enjoyed reading of the manuscript. Well designed and documented studies, well written paper, topic highly relevant. However, I feel that one important issue needs to be solved: does the GFD in Study 1 and Study 2 include gluten-free oats or not? Gluten-free oats may increase nutritional value of a GFD via improved amino acid, fiber, mineral, and vitamin balance. In Study 2, there is necessary to emphasize inclusion/exclucion of gluten-free oats.
Response 1: That is a very good point, thank you for bringing it to our attention. We did include gluten free oats in study 1. We added a line specifying this in the methods. However, the studies used in our meta-analysis did not specify whether or not oats were included in a gluten free diet. We added a line stating this as a limitation in the discussion.
Point 2: In Study 1, it would be great to add a sub-study that calculates intake of fiber and other mentioned nutrients, minerals and vitamins.
Response 2: We examined the difference in fiber and other shortfall nutrients between a GF and GC diet in study 1. The results are displayed in table 2.
Thank you for your review!

Round 2
Reviewer 2 Report
no further comments